# Prevalence and Characterization of Quinolone-Resistance Determinants in *Escherichia coli* Isolated from Food-Producing Animals and Animal-Derived Food in the Philippines

**DOI:** 10.3390/antibiotics10040413

**Published:** 2021-04-09

**Authors:** Lawrence Belotindos, Marvin Villanueva, Joel Miguel, Precious Bwalya, Tetsuya Harada, Ryuji Kawahara, Chie Nakajima, Claro Mingala, Yasuhiko Suzuki

**Affiliations:** 1Division of Bioresources, International Institute for Zoonosis Control Hokkaido University, Sapporo 001-0020, Japan; lawrence@czc.hokudai.ac.jp (L.B.); preciousbwalya@czc.hokudai.ac.jp (P.B.); cnakajim@czc.hokudai.ac.jp (C.N.); 2Biosafety and Environment Section, Philippine Carabao Center, Science City of Muñoz, Nueva Ecija 3120, Philippines; mavillanueva_vet@hotmail.com (M.V.); cnmingala@hotmail.com (C.M.); 3Livestock Biotechnology Center, Philippine Carabao Center, Science City of Muñoz, Nueva Ecija 3120, Philippines; joel.miguel.6194@gmail.com; 4Division of Microbiology, Osaka Institute of Public Health, Osaka 537-0025, Japan; kawahagi.9510@gmail.com (R.K.); harada@iph.osaka.jp (T.H.); 5International Collaboration Unit, International Institute for Zoonosis Control, Hokkaido University, Sapporo 001-0021, Japan

**Keywords:** *Escherichia coli*, quinolone, QRDR, PMQR, the Philippines

## Abstract

Antimicrobial resistance to quinolones, which constitutes a threat to public health, has been increasing worldwide. In this study, we investigated the prevalence of quinolone-resistant determinants in *Escherichia coli* not susceptible to quinolones and isolated from food-producing animals and food derived from them, in the Philippines. A total of 791 *E. coli* strains were isolated in 56.4% of 601 beef, chicken, pork, egg, and milk samples, as well as environmental, cloacal, and rectal swab-collected samples from supermarkets, open markets, abattoirs, and poultry, swine, and buffalo farms. Using the disc diffusion method, it was determined that 78.6% and 55.4% of the isolates were resistant to at least one antimicrobial and multiple drugs, respectively. In 141 isolates not susceptible to quinolones, 115 (81.6%) harbored quinolone-resistant determinants and had mutations predominantly in the quinolone-resistance determining regions (QRDRs) of *gyrA* and *parC*. Plasmid-mediated, quinolone resistance (PMQR) and Qnr family (*qnrA1*, *qnrB4*, and *qnrS1*) genes were detected in all isolates. Forty-eight sequence types were identified in isolates harboring mutations in QRDR and/or PMQR genes by multilocus sequence typing analysis. Moreover, 26 isolates harboring mutations in QRDR and/or PMQR genes belonged mostly to phylogroup B1 and Enteroaggregative *E. coli*. In conclusion, a high prevalence of *E. coli* was found in food-producing animals and products derived from them, which could potentially spread high-risk clones harboring quinolone-resistance determinants.

## 1. Introduction

Antimicrobials are necessary tools to fight diseases that create an economic burden, while at the same time contributing to health, welfare, food safety, and food security for both animals and humans [1]. The overuse of antimicrobials has led to the emergence of antimicrobial-resistant microorganisms in food-producing animals and those products derived from them such as meat, eggs, and milk. Consuming or being in contact with food containing antimicrobial-resistant microorganisms can cause the development of foodborne diseases that are difficult to treat [2]. In 2010 alone, over 400,000 people died due to foodborne diseases, which were caused by microorganisms such as bacteria, with over one-third of these deaths being children under the age of five [3]. Therefore, foodborne diseases and the emergence of antimicrobial resistance are both public health concerns that need to be addressed on a global scale.

Quinolones are essential antimicrobials to treat bacterial infections in both animals and humans. Due to the rapid development of and increase in quinolone-resistant strains, the World Health Organization (WHO) has recommended reducing their use in livestock [4,5,6,7]. Quinolones prevent the activity of DNA gyrase and topoisomerase IV, which results in chromosomal fragmentation and death of bacteria [8]. Quinolone resistance in bacteria is acquired by the presence of one or more target-site mutations at quinolone-binding sites known as quinolone-resistance determining regions (QRDRs) in genes encoding DNA topoisomerases (*gyrA*, *gyrB*, *parC*, and *parE*). The development of mutations alter the drug-binding affinity with target enzymes [9]. Although they confer a low-level resistance to quinolones such as *qnr*, *acc(6′)-lb-cr*, and *qepA*, the recent discovery of plasmid-mediated quinolone resistance (PMQR) genes has aggravated the concern of health organizations. Indeed, for example, gene *qnr* encodes the pentapeptide protein competing with quinolones, *acc(6′)-lb-cr* encodes the mutated aminoglycoside acetyltransferase (which can modify ciprofloxacin), and *qepA* encodes an efflux pump protein. Moreover, these genes can be spread horizontally across Enterobacteriaceae and positively contribute to the development of chromosome-encoded quinolone resistance mechanisms [10,11,12].

The detection of quinolone/fluoroquinolone-resistant bacteria has been increasing in the Philippines. For instance, the prevalence rates of nalidixic acid and ciprofloxacin resistance in strains isolated from food-producing animals and food derived from them were 10–97.5% and 5–88.4%, respectively [13,14,15,16,17]. In environmental samples from soil and agricultural irrigation water, the detected resistance rate against nalidixic acid and ciprofloxacin was up to 35.4% and 6.8%, respectively [18,19]. In clinical settings, increasing resistance rates in *E. coli* (ciprofloxacin, 39%) and nontyphoidal *Salmonella* (ciprofloxacin, 10.3%) have been observed [20]. Previous studies focused on the phenotypic characterization of the resistance determinants of bacteria to some extent. However, those studies mostly scrutinized the extended beta-lactamase (ESBL) resistance mechanism of bacteria [16,17].

Elucidating the mechanism underlying the acquisition of antimicrobial-resistant genes would enable a deeper understanding of transmission that is crucial for effective infection control. The data on the quinolone-resistance acquisition mechanism of *E. coli* in food-producing animals and products derived from them are limited, despite *E. coli* being listed as a priority pathogen in the Philippines by the WHO. Therefore, the present study aimed to investigate the quinolone-resistance determinants in *E. coli* isolated from food-producing animals and their food products in the Philippines. Furthermore, to elucidate the dissemination of high-risk clones, in the present work, the relationship between quinolone-resistant isolates was analyzed.

## 2. Results

### 2.1. Prevalence of E. coli in Samples

Of the 601 samples collected, 339 (56.4%) were positive for *E. coli*, ranging from 47.8% to 87.2% in meat samples. The isolation rates of *E. coli* were 53.3% in cloacal swab samples, 64.4% in rectal swab samples, and 73.3% in environmental swab samples (4/4 from abattoirs, 23/31 from swine farms, and 6/10 from poultry farms). In contrast, only one egg sample (2%, 1/50) was positive for *E. coli*, and no *E. coli* isolate was found in milk samples (Table 1).

### 2.2. Antimicrobial Susceptibility

In total, 791 *E. coli* isolates were detected in samples. The antimicrobial susceptibility of isolates is shown in Table 2 and Appendix A. Of the tested antimicrobials, resistance rates to tetracycline, ampicillin, sulfamethoxazole-trimethoprim, chloramphenicol, streptomycin, nalidixic acid, kanamycin, and ciprofloxacin were 57.5%, 56.6%, 43.7%, 35.9%, 30.0%, 25.3%, 14.7%, and 11.6%, respectively. Resistance to colistin, carbapenems (imipenem and meropenem), and cephems (cefoxitin, cefotaxime, ceftazidime, and cefepime) was detected in less than 10% of the isolates (Table 2). Overall, 21.45% of the isolates were pan-susceptible, 78.6% were resistant to at least one antimicrobial agent, and 55.4% were multidrug-resistant (i.e., resistance to at least one agent in three or more antimicrobial categories) (Table 3). The multidrug-resistant rates were 27.2%, 48.4%, and 67.4% in beef, pork, and chicken samples, respectively. Notably, the multidrug-resistant rate of isolates in chicken samples was significantly higher than that in other meat samples (*p* < 0.05). By contrast, while multidrug-resistant rates in swab samples were high, no significant differences were observed (Table 3).

### 2.3. QRDR and PMQR Determinant Analysis

In 141 isolates not susceptible to quinolones, 46.8% had an amino-acid substitution in the QRDR of *gyrA*, *gyrB*, *parC*, and *parE*; 19.8% harbored PMQR genes, whereas 14.9% had both of them. The predominant amino-acid substitution in the QRDR of *GyrA* was serine to leucine at codon 83 (Ser83 Leu) (95.4%, 84/87) and aspartic acid to asparagine (Asp87 Asn) (67.8%, 59/87) or tyrosine (Asp87 Tyr) (3.4%, 3/87) at codon 87. In *GyrB*, serine to aspartic acid at codon 492 (Ser492 Asn) (11.5%, 10/87) was the predominant amino-acid substitution. In ParC, the most frequent substitutions were serine to isoleucine (Ser80 Ile) (70.1% 61/87) or arginine (Ser80 Arg) (1.1%, 1/87) at codon 80 and glutamic acid to glycine at codon 84 (Glu84 Gly) (10.3%, 9/87). In *ParE*, the predominant amino-acid substitutions were serine to alanine at codon 458 (Ser452 Ala) (18.4%, 16/87) and isoleucine to phenylalanine at codon 464 (Ile464 Phe) (2.3%, 2/87). Amino-acid substitutions at codon 83 and 87 in *gyrA*, along with substitutions at codon 80 in *parC*, were the most frequent substitution patterns (31/141) (Figure 1). Isolates carrying double amino-acid substitutions in *GyrA* plus a single or double amino-acid substitution in other QRDR genes demonstrated a high-level quinolone resistance (Table 4).

PMQR and qnr-family genes were detected in 49 (34.8%) isolates. The most frequent qnr was qnrS1 found in 31 isolates (63.2%). In one of these isolates, qnrS1 coexisted with qnrA1, while three isolates had additional amino-acid substitutions in *gyrA*. The second most frequent qnr was qnrB4, which was detected in 18 isolates (36.7%). All qnrB4-positive isolates had an additional amino-acid substitution in QRDR, with the exception of *parE* (Figure 1). All PMQR-positive isolates displayed a variable resistance to quinolones (minimum inhibitory concentration (MIC) of ciprofloxacin <0.03 to >32 μg/mL and MIC of nalidixic acid 8 to >128 μg/mL). Notably, the presence of both of qnrA1 and qnrS1 exhibited a high-level quinolone resistance (MIC of ciprofloxacin >32 μg/mL and MIC of nalidixic acid >128 μg/mL). The other PMQR genes, namely, *qepA* and *acc(6′)-Ib-cr*, were not detected in any of the tested isolates (Figure 1 and Table 4). In total, 115 (81.6%) isolates had quinolone-resistance determinants, whereas, in 26 (18.4%), no quinolone-resistant determinants were observed.

### 2.4. Multilocus Sequence Typing and Phylogenetic Group Analysis

In 115 isolates harboring mutations in QRDR and PMQR determinants, 46 unique sequence types (STs) were identified. Seventy-eight (67.8%) of the isolates were clustered into 13 clonal complexes (CCs), while 37 (32.2%) isolates were singletons. The most common CCs were CC155 (*n* = 16), followed by CC101 (*n* = 15), CC469 (*n* = 10), CC10, and CC206 (*n* = 9). Overall, ST155 (*n* = 13, 11.3%) was the most frequent ST, followed by ST162 (*n* = 11, 9.6%), ST359 (*n* = 9, 7.8%), and ST354 (*n* = 8, 7%). Thirteen STs were found in more than one sample type (Figure 1 and Figure 2).

The majority of the isolates were assigned to phylogenetic group B1 (55.7%) (*p* < 0.05), followed by group A (28.7%) and group D (15.7%). None of the isolates belonged to phylogenetic group B2 (Figure 1). While isolates with amino-acid substitutions in QRDRs were significantly associated with phylogenetic group B1 (*p* < 0.05), isolates harboring PMQR genes were significantly associated with phylogenetic group A (*p* < 0.05). In contrast, isolates that harbored both QRDR and PMQR genes were distributed evenly among phylogenetic groups (A, B1, and D) (Figure 3).

### 2.5. Prevalence of Virulence Genes

Twenty-six *E. coli* isolates harboring mutations in QRDR and/or PMQR genes carried one virulence gene, *astA*, encoding the enteroaggregative heat-stable enterotoxin 1 (EAST1) of Enteroaggregative *E. coli*. The remaining 89 (77.4%) isolates carried no virulence genes (Figure 1).

## 3. Discussion

This is the first report investigating the quinolone-resistance determinants and molecular characteristics of isolates not susceptible to quinolones (intermediate and/or resistant to nalidixic acid and/or ciprofloxacin, using the disc diffusion method) in the Philippines.

*E. coli* is a common bacterium, but some of its strains can cause diseases in both animals and humans. Indeed, pathogenic *E. coli* is usually a food contaminant that can cause severe public health problems. Furthermore, *E. coli* serves as a reservoir for drug-resistant genes that can be horizontally transferred to other pathogenic bacteria [21]. In the present study, 54.6% of the samples were contaminated with *E. coli*, with the highest prevalence being observed in chicken meat (70.6–82.7%). The high-level contamination of *E. coli* in chicken was consistent with that reported by previous studies conducted in the Philippines [15,22], China [23], and Bangladesh [24]. Contamination of meat samples with pathogenic *E. coli* usually indicates poor hygiene during slaughter and handling, and unsuitable storage after slaughter [22,23]. Another source of contamination is meat supplied from unauthorized abattoirs (e.g., no proper hygiene inspections by authorities), which is then sold at local retail meat shops. Contamination with pathogenic bacteria jeopardizes food safety and human health. For example, according to a Philippines foodborne disease outbreaks report (2005–2018), 14.35% of infection cases were associated with animal-derived food [25]. Therefore, monitoring the food production process and the implementation of food hygiene practices is essential and, hence, improved practices and management in slaughterhouses must be implemented.

Among the antimicrobials used in the present study, resistance to tetracycline, ampicillin, and sulfamethoxazole/trimethoprim was frequently observed. The resistance pattern detected in the present work was concordant with previous reports in the Philippines [13,14,15]. These antimicrobials are commonly used in livestock and poultry in the Philippines [26]. The extent and pattern of antimicrobial-resistant rates seen in *E. coli* isolated from meat samples were very similar to those in isolates from farms. Therefore, a possible link could be established to antimicrobial usage at farm level, which leads to a high number of resistant *E. coli* in food products. Furthermore, indirect antimicrobial ingestion through animal-derived food consumption is projected to increase due to an increased demand in animal products in the Philippines [27]. To meet the demand, to maintain the health of animals, and to increase their productivity, there has been a shift in production systems from backyards to intensive production systems that rely more on antimicrobials. Notably, a high resistance to chloramphenicol has been found despite the practice being prohibited [28], and even resistance to carbapenems, which is not commonly used in livestock animals in the Philippines, was observed in the present study. The resistance to these two antimicrobials may be due to cross- or co-resistance against the same or other antimicrobial classes, which were possibly used illegally [15]. Therefore, there is a need to fully enforce the laws regarding the usage of these antimicrobials and to monitor the presence of resistant bacteria that may pose a risk to food safety and public health.

Overall, 55.4% of the *E. coli* isolates were multidrug-resistant. The *E. coli* multidrug-resistant rates in animals and animal-derived food in this study were lower compared with those of previous studies in the Philippines (70–95%) [13,14,15] and neighboring countries such as Thailand and Cambodia (75.3%) [29]. Yet, it was slightly higher than in food samples from Myanmar (50%) [24] and in human clinical samples (46% in blood samples) [30] in the Philippines. These discrepancies may be explained by the fact that previous studies focused on specific resistance phenotypes [12], with differences in target samples, geographical dominant strains, and numbers of samples tested.

Quinolone and fluoroquinolones are the drugs of choice for human foodborne and other infections caused by *Salmonella* and *E. coli* [31,32]. They are also used as prophylactics and for treatment of chronic respiratory diseases, skin and soft tissue infections, urinary tract infections, enteritis, and mastitis in animals [33]. However, there is an increasing presence of resistance to quinolone and fluoroquinolone in bacteria that threaten their efficacy. They are, therefore, classified as high-priority, critical drugs, and it has been recommended to reduce their use in food animals [34]. Some countries (United States of America (USA), Finland, the Netherlands, and Australia) have already reduced or banned altogether the usage of fluoroquinolones in food-producing animals [4,5,35]. In Southeast Asia, however, including the Philippines, they are still employed in animal production. In the present study, high numbers of *E. coli* not susceptible to quinolones/fluoroquinolones were observed in chicken, followed by swine-associated samples, which were similar to those detected by recent but unrelated reports in the Philippines [16,17]. These findings may correspond with the use of quinolone/fluoroquinolone in poultry and swine production in the Philippines. There are at least three to four quinolones/fluoroquinolones, primarily enrofloxacin, used in poultry and swine (backyard/commercial) farms as growth-promoting, therapeutic, or prophylactic agents [26]. Therefore, to minimize the development/acquisition of resistance to these antimicrobials, it is necessary to establish a proper monitoring of the usage of antimicrobials in the Philippines.

The prevalence of mutations in QRDR and PMQR genes in *E. coli* isolates from humans, animals, and the environment has been found in many countries [35,36,37,38,39,40]. However, limited data are available on the presence of chromosomal mutations in the QRDR and PMQR genes conferring resistance to quinolones in samples from food-producing animals in the Philippines. In the present study, we found that isolates not susceptible to quinolones had predominantly double amino-acid substitutions in *GyrA* (Ser83 to Leu; Asp87 to Asn) and an amino-acid substitution in *ParC* (Ser80 to Ile). This QRDR mutation pattern has been frequently observed in fluoroquinolone-resistant *E. coli* clinical isolates in the Philippines and elsewhere [38]. In the previous studies, a single mutation of Ser83 in *gyrA* was enough to cause a high-level resistance to nalidixic acid and decreased susceptibility to fluoroquinolones. However, with an additional mutation in *gyrA* and *parC*, it is a stepwise event that caused a high-level resistance to nalidixic acid and ciprofloxacin [8,9,11,33]. This evidence correlates with our results showing the observed isolates having these mutations, as well as high resistance to nalidixic acid (MIC > 128 μg/mL) and full resistance to ciprofloxacin (MIC 4 to >32 μg/mL).

The qnr are known to protect DNA gyrase against the effect of quinolones [33,41,42]. We found that 34.8% of isolates not susceptible to quinolones harbored qnr in the present study. Most of the qnr-harboring isolates were moderately to fully resistant to nalidixic acid (MIC 8 to >128 μg/mL) and susceptible to less susceptible to ciprofloxacin (MIC 0.25 to 2 μg/mL). In addition, we observed a coexistence of qnrA1 and qnrS1 in an isolate with high MIC (MIC of ciprofloxacin >32 μg/mL and MIC of nalidixic acid >128 μg/mL). The identification of qnrA1 in the present work is the first reported in the Philippines. The MICs of ciprofloxacin for *E. coli* harboring qnrA1 were reported to be 0.12–0.25 μg/mL [23,40]. In the present study, isolates harboring qnrS1 alone had 0.25–2 μg/mL MIC of ciprofloxacin (Table 4). These findings seem to indicate that the combination of qnrA1 and qnrS1 did not compete for binding in gyrases and had a synergistic or additive effect with the MIC, although the actual mechanism remains unclear. Coexistence of qnrA and qnrS has also been found in *Enterobacter cloacae* showing higher (2–8-fold) MICs of quinolones than strains harboring only qnrA, seemingly indicating that both genes had an additive effect when conferring quinolone resistance [41]. In contrast, past work showed the coexistence of qnr in one isolate, which tended to have the same resistance activity as that of a single one [42]. In the present study, we identified qnr (qnrA1, qnrB4, and qnrS1) in 28 isolates not susceptible to quinolones. Gene qnrS1, in particular, was observed most frequently, which is in agreement with work reported elsewhere [7,12,23,36]. Again, this result seems to indicate that qnrS1 is the predominant gene in food-producing animals and in their food products. Within the qnr family, qnrB is the most commonly observed [43]. In the present study, however, qnrB4 was the only one observed among qnrB alleles, but it was less predominant and mostly detected in chicken meat samples, similar to data reported in Korea [6]. Gene qnrB4 was also found in extended-spectrum beta-lactamase-producing Enterobacteriaceae from clinical samples in the Philippines [44,45]. PMQR genes are often found in plasmids with other antibiotic-resistant genes and can be horizontally transferred to other bacteria even without antibiotic exposure, thus becoming a source of quinolone resistance during infection in humans [41,42]. However, in the present study, a plasmid carrying the gene qnr was not implemented, whether polymerase chain reaction assay, whole genome sequencing, or S1 nuclease pulsed field gel electrophoresis; hence, this should be further elucidated.

Molecular typing studies of *E. coli* are of limited scope in the Philippines. In the present study, ST155, ST162, ST359, and ST354 were predominantly detected in *E. coli* isolates having mutations in QRDR and/or PMQR genes. These clones were previously reported to be associated with human and animal infections [46,47,48]. In addition, *E. coli* ST38 and ST155 are considered high-risk clones that disperse antibiotic resistance on a global scale. These clones have acquired adaptive traits that increase pathogenicity to colonize, spread, and thrive in a variety of niches [49,50]. Clones ST10, ST48, ST162, and ST206, detected in the present work in beef, chicken, pork, and cloacal swab samples, were previously associated with carbapenemase-producing *E. coli*, which was isolated from hospital sewage and river samples in the Philippines, and they showed a reduced to high level of resistance to levofloxacin [51]. Moreover, ST10 and ST117 have been associated with emerging extraintestinal pathogenic *E. coli* (ExPEC) lineages that cause infections in humans [52]. In the present study, all ST10 isolates were multidrug-resistant and mostly carried qnrs and a virulence gene, whereas ST117 was recovered from two different samples (chicken and pork), with both isolates being multidrug-resistant with mutations in QRDRs. ST117 is a well-recognized avian pathogenic *E. coli* with zoonotic potential [53]. In addition, in the present work, 13 STs were observed in multiple sources, seemingly indicating a possible interspecies transmission that potentially poses a risk to humans via direct contact with food-producing animals and/or consumption of contaminated animal-derived food. Therefore, to prevent possible infection outbreaks, there is a need to screen for the presence of these pathogenic STs in food-producing animal and their food products.

In the present study, the majority of the isolates belonged to phylogenetic groups A, B1, and D. These results are similar to those reported in previous work on *E. coli* not susceptible to quinolones [36]. Although *E. coli* belonging to phylogenetic groups A and B1 are classified as environmental and commensal *E. coli*, strains that belong to group D are classified as potential extraintestinal pathogenic strains [54]. In the present work, it was found that 15.7% of *E. coli* isolates harboring mutations in QRDR and/or PMQR genes belonged to phylogenetic group D. These isolates include *E. coli* clonal lineages (ST38, ST117, and ST354), which are human-associated, fluoroquinolone-resistant lineages that cause extraintestinal infection [53,55]. These findings indicate that these isolates may also carry pathogenic characteristics of ExPEC strains. Furthermore, 22.6% of the *E. coli* isolates harboring mutations in QRDR and/or PMQR genes in the present study harbored *astA*, encoding EAST1. It can be, therefore, hypothesized that *astA* could elicit a cyclic guanosine monophosphate increase, leading to the loss of electrolytes and water from the epithelial intestinal cells, similar to the heat-stable enterotoxin mechanism [56]. Past epidemiological studies have shown that this gene was also present in other major pathogenic *E. coli* and even in a commensal strain [57]. In addition, the association of *astA* with foodborne outbreaks has been reported in many countries including Japan, Chile, Thailand, and Kenya [58,59,60,61]. The presence of EAST1 in food-producing animals and their food products is of great concern due to the notion that the spread of enteroaggregative *E. coli* strains may increase food-borne diarrheic infections. Therefore, these isolates have the potential to cause human infection given the ideal conditions, whereby it is difficult to render treatment due to predominantly multidrug-resistant isolates carrying quinolone-resistance determinants.

## 4. Materials and Methods

### 4.1. Sample Collection, Bacterial Enrichment, and Isolation

A total of 601 samples (beef, chicken, and pork samples from supermarkets, open-air markets, and abattoirs, milk samples from dairy buffalo farms, cloacal swabs and eggs from poultry farms, rectal swabs from pig farms, and environmental swabs from abattoirs and poultry and swine farms) were collected in the Philippines from November 2017 to July 2018. Sampling methods were based on convenience and efficiency. Bacterial isolation was conducted using a culture-based method. Briefly, 25 g or an equal volume of each sample was diluted 1:10 in Brain Heart Infusion broth (Nissu Pharm Co. Ltd., Tokyo, Japan) and homogenized using a sample blender (Bag Homogenizer BH-W, AS ONE Corp., Osaka, Japan). Next, the homogenates were incubated for 18–22 h at 44 °C. After incubation, the homogenates were streaked on plates with MacConkey agar (Becton Dickinson Co., Ltd., Franklin Lakes, NJ, USA) and incubated for 18–22 h at 37 °C. Five presumptive *E. coli* colonies (color: brick-red) were selected and identified using a standard biochemical test as previously described [29]. Confirmed *E. coli* isolates were stored at −20 °C in Luria–Bertani broth (Merck, Darmstadt, Germany) containing glycerol 50% *v*/*v* (Difco Laboratories, Inc., Detroit, MI, USA).

### 4.2. Antimicrobial Susceptibility Testing

All confirmed *E. coli* isolates were subjected to antimicrobial susceptibility testing using the Kirby–Bauer disc diffusion method according to the Clinical and Laboratory Standards Institute [62] standard protocol, using commercially available antibiotic discs (Becton Dickinson Co., Ltd.). Seventeen antimicrobial agents were used in the present study: kanamycin (30 μg), gentamicin (10 μg), streptomycin (10 μg), cefoxitin (30 μg), cefotaxime (30 μg), cefepime (30 μg), ceftazidime (30 μg), imipenem (10 μg), meropenem (10 μg), nalidixic acid (30 μg), ciprofloxacin (5 μg), ampicillin (10 μg), amoxicillin/clavulanic acid (20 μg/10 μg), chloramphenicol (30 μg), tetracycline (30 μg), sulfamethoxazole/trimethoprim (23.75 μg/1.25 μg), and colistin (10 μg). The results were classified as susceptible, intermediate, or resistant (Appendix A) as per the diameter of the zone of inhibition, using Clinical and Laboratory Standards Institute breakpoints [62]. An isolate was considered multidrug-resistant if it was resistant to at least one agent from three or more antimicrobial categories. In addition, the MICs of nalidixic acid and ciprofloxacin were determined using broth microdilution [63] for isolates harboring mutations in QRDR and/or PMQR genes, and the results interpreted according to the Clinical and Laboratory Standards Institute breakpoints [64]. *E. coli* ATCC 25,922 was used as the control strain.

### 4.3. Detection of Quinolone-Resistance Determinants

Using the disc diffusion method, a total of 141 *E. coli* isolates not susceptible to quinolones, intermediate and/or resistant to nalidixic acid and/or ciprofloxacin, were selected by screening for quinolone-resistance determinants. Genomic DNA was extracted using the boiling method. Briefly, the bacterial colonies were suspended in 500 μL of TE buffer (Tris-HCl (10 mM), Ethylenediaminetetraacetic acid (EDTA) (1 mM)) in microcentrifuge tubes and subjected to 15 min of boiling. Immediately after boiling, the microcentrifuge tubes were centrifuged for 10 min at 12,000× *g* at room temperature. The supernatant containing DNA (100 µL) was transferred to new sterile microcentrifuge tubes and used for the polymerase chain reaction (PCR) analysis. PMQR determinants (*qnrA*, *qnrB*, *qnrS*, *qepA*, and *acc(6′)-lb-cr*) and amino-acid substitutions in the QRDRs of *GyrA*, *GyrB*, *ParC*, and *ParE* were determined by PCR and sequencing as previously described [23,65,66,67,68,69,70]. The PCR mixture (20 µL) contained 1.25 U of Taq DNA polymerase (Promega, Madison, WI, USA), 1× PCR buffer (Promega), 0.2 mM of each dNTP (Takara Bio Inc., Shiga, Japan), 2.5 mM MgCl_2_ (Promega), 1 µM of each primer, and 1 µL of DNA template. The PCR conditions for QRDR were as follows: for *GyrA* and *GyrB*, denaturation at 96 °C for 1 min followed by 35 cycles of denaturation at 96 °C for 10 s, annealing at 52 °C for 10 s, an extension at 72 °C for 30 s, and a final extension at 72 °C for 5 min; for *ParC* and *ParE*, denaturation at 94 °C for 3 min followed by 35 cycles of denaturation at 94 °C for 30 s, annealing at 55 °C for 30 s, an extension at 72 °C for 30 s, and a final extension at 72 °C for 5 min. The PCR conditions for PMQR were as follows: for *qnrB*, denaturation at 95 °C for 5 min followed by 35 cycles of denaturation at 94 °C for 30 s, annealing at 56 °C for 40 s, an extension at 72 °C for 1 min, and a final extension at 72 °C for 10 min; denaturation at 95 °C for 5 min followed by 35 cycles of denaturation at 94 °C for 45 s, annealing at 51 °C (for *qepA*), 53 °C (for *qnrA* and *qnrS*), 55 °C (for *acc(6′)-lb-cr*) for 45 s, an extension at 72 °C for 1 min, and a final extension at 72 °C for 5 min.

PCR products were purified using ExoSAP^®^ IT (Thermo Fisher Scientific Co., Ltd., MA, USA) and subjected for sequencing using a BigDye^®^ ver. 3.1 Terminator Cycle Sequencing Kit (Thermo Fisher Scientific Co., Ltd.) in an ABI 3500 xL Genetic Analyzer (Thermo Fisher Scientific Co., Ltd.). The obtained sequences were confirmed using data from the National Center for Biotechnology Information website (www.ncbi.nlm.nih.gov/BLAST/ (accessed on 28 February 2021)). Furthermore, inferred amino-acid sequences of QRDR-encoding genes were aligned with the corresponding regions of *E. coli* K-12 (GenBank accession no. AL513382.1) as a reference strain using the ClustalW program by MEGA v7.0.21.

### 4.4. Multilocus Sequence Typing Analysis

Genotyping of all *E. coli* isolates harboring mutations in QRDR and/or PMQR genes was conducted as per the multilocus sequence typing protocol for *E. coli* [71]. Seven housekeeping genes, namely, *adk*, *fumC*, *gyrB*, *icd*, *mdh*, *purA*, and *recA*, were amplified using the recommended primers. The PCR mixture was the same as that mentioned in Section 4.3. The PCR conditions were as follows: denaturation at 95 °C for 5 min, followed by 30 cycles consisting of denaturation of 95 °C for 1 min, annealing at 54 °C (for *adk*, *fumC*, *icd*, and *purA*), 58 °C (for recA), 60 °C (for *mdh*) for 1 min, an extension at 72 °C for 2 min, and a final extension at 72 °C for 5 min. The amplified products were purified using ExoSAP^®^ IT (Thermo Fisher Scientific Co., Ltd.) and subjected to bidirectional sequencing using the BigDye^®^ ver. 3.1 Terminator Cycle Sequencing Kit (Thermo Fisher Scientific Co., Ltd.) in the ABI 3500 xL Genetic Analyzer (Thermo Fisher Scientific Co., Ltd.). To determine the respective alleles, sequence types, clonal complexes, and singleton assignments, the obtained sequences were submitted to the multilocus sequence typing database (https://pubmlst.org/escherichia/andenterobase.warwick.ac.uk (accessed on 28 February 2021)). A minimum spanning tree/unweighted pair group method with an arithmetic mean (UPGMA) was generated following the cluster analysis of the multilocus sequence typing allelic profiles of the isolates using BioNumerics 6.6 software (Applied Maths, Sint-Martens-Latem, Belgium).

### 4.5. Detection of Virulence Genes

Major virulence determinants associated with major *E. coli* pathotypes were determined in all *E. coli* isolates harboring mutations in QRDR and/or PMQR genes. Pathotypes were identified according to the presence of specific virulence genes (VGs): Shiga toxin-producing *E. coli* (*stx1* and/or *stx2*, *eae*), typical/atypical Enteropathogenic *E. coli* (*eaeA*, *bfpA*/*eaeA*), Enterotoxigenic *E. coli* (*elt*, *STp*, *STh*, and *astA*), Enteroinvasive *E. coli* (*invE*) and, Enteroaggregative *E. coli* (*astA*, *aggR*), as described previously [72]. The PCR mixture was the same as that mentioned in Section 4.3. The PCR conditions were as follows: denaturation at 94 °C for 2 min, followed by 30 cycles consisting of denaturation of 94 °C for 1 min, annealing at 52 °C (for *eae*), 55 °C (for *aggR*, *elt*, *STp*, *STh*, *invE*, *astA*, and *recA*), 56 °C (for *stx2*), and 58 °C (for *stx1*) for 1 min, an extension at 72 °C for 2 min, and a final extension at 72 °C for 10 min. PCR amplicons were visualized on 1.5% agarose gels stained with GelRed (Biotium, Inc., CA, USA). After the gel electrophoresis, images of the PCR amplicons were captured using Printgraph Classic (ATTO Corp., Tokyo, Japan). Control DNA (*E. coli* O157:H7 for *stx1* and *stx2*, *E. coli* O125:H45 for *eae* and *bfpA*, *E. coli* O6:H16 for *elt*, *STh*, *E. coli* O169:H41 for *STp*, *E. coli* O11: H30 for *aggR*, *E. coli* O25:HNM for *elt*, and *Shigella flexneri* 2a for *invE*) kindly provided by the Division of Microbiology, the Osaka Institute of Public Health, Japan was used in each PCR experiment.

### 4.6. Phylogenetic Group Analysis

All *E. coli* isolates harboring mutations in QRDR and/or PMQR genes were assigned to phylogenetic groups (A, B1, B2, or D) on the basis of the presence or absence of genes *chuA* and *yjaA* and the DNA fragment tspE4 C2 by triplex-PCR, as previously described [54]. The PCR mixture was the same as that mentioned in Section 4.3. The PCR conditions were as follows: denaturation at 94 °C for 4 min, followed by 30 cycles of denaturation at 94 °C for 5 s and annealing at 59 °C for 10 s, and a final extension at 72 °C for 5 min. PCR amplicons were visualized using 1.5% agarose gels stained with GelRed (Biotium, Inc., CA, USA), and their images were captured using Printgraph Classic (ATTO Corp.).

### 4.7. Data Analysis

The data were descriptively analyzed using Statistical Package for Social Sciences (SPSS) software version 22.0 (IBM Corp., Armonk, NY, USA). Differences in proportions were compared using a χ^2^ test or Fisher’s exact test. All the tests were analyzed with a 95% confidence interval. A *p*-value less than 0.05 was considered statistically significant.

## 5. Conclusions

This is the first study to focus on the molecular characteristics of *E. coli* not susceptible to quinolone found in food-producing animals and their food products in the Philippines. The quinolone-resistance determinants of *E. coli* in the Philippines were found to be mediated predominantly by amino-acid substitutions in QRDRs of *GyrA* and *ParC*. In addition, a high prevalence of PMQR genes was detected, which raises concerns about the broad dissemination of drug-resistant strains. Lastly, high ST diversity within *E. coli* isolates harboring mutations in QRDR and/or PMQR genes indicated that the spread of quinolone resistance strains in the Philippines is not dependent on a specific clone. According to our results, we recommend that the hygiene laws for animal slaughter and food handling be enforced in the Philippines, as a high multidrug-resistant *E. coli* contamination rate was observed not only in abattoirs, but also in animal-derived food, from both supermarkets and open markets. Furthermore, to minimize the emergence and spread of quinolone-resistant *E. coli*, we recommend the implementation of a strict monitoring of antimicrobial use and the restriction of quinolone usage for therapeutic and farming purposes.

## Figures and Tables

**Figure 1 antibiotics-10-00413-f001:**
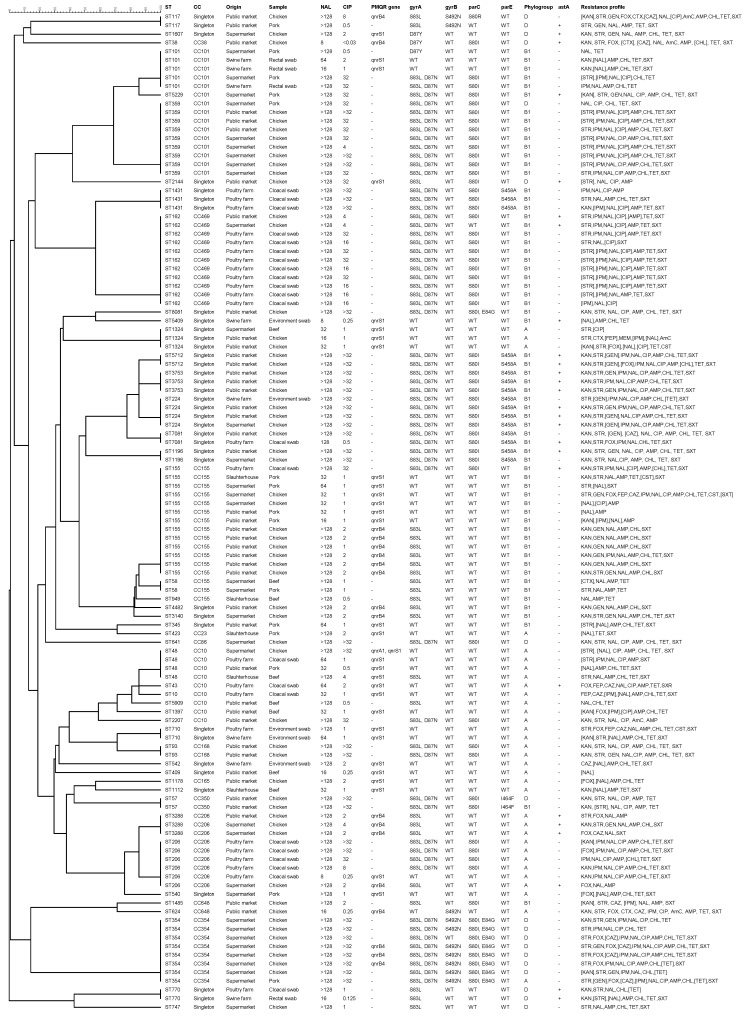
Dendrogram showing the relationship between 115 plasmid-mediated quinolone resistance (PMQR)/quinolone-resistance determining region (QRDR) gene-harboring *E. coli* strains isolated from beef, chicken, pork, and cloacal, rectal, and environmental swab samples, based on the multilocus sequence typing (MLST) allele profile including information about sequence type (ST), clonal complex (CC), minimum inhibitory concentration (MIC) of nalidixic acid (NAL)/ciprofloxacin (CIP), gyrase/topoisomerase substitutions, phylogroup, virulence gene (*astA*), and phenotypic resistance profile according to the Kirby–Bauer disc diffusion method. KAN—kanamycin; STR—streptomycin; GEN—gentamicin; FOX—cefoxitin; CTX—cefotaxime; FEP—cefepime; CAZ—ceftazidime; IPM—imipenem; MEM—meropenem; AMP—ampicillin; AmC—amoxicillin/clavulanic acid; CHL—chloramphenicol; TET—tetracycline; STX—trimethoprim/sulfamethoxazole; CST—colistin. Antimicrobial agents inside bracket indicate intermediate resistance. (+)—positive; (−)—negative; WT—wild type.

**Figure 2 antibiotics-10-00413-f002:**
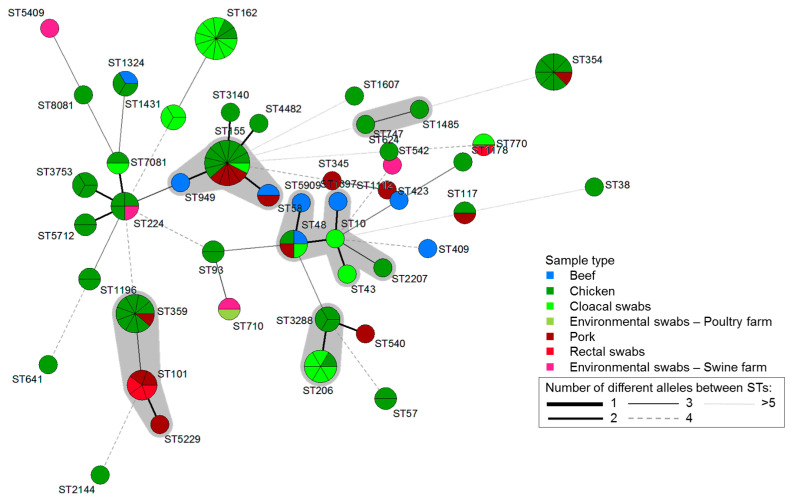
Minimum spanning tree based on multilocus sequence typing alleles of 115 QRDR/PMQR-harboring *E. coli* isolates from beef, chicken, pork, cloacal swabs, rectal swabs, and environmental swabs. Each circle corresponds to an individual sequence type (ST), and the circle size indicates the number of isolates assigned to the same ST. The color of the circle denotes the sample type. The connecting lines (solid and dashed) between circles denote allelic variations between STs, and the gray shadowing indicates STs belonging to the same clonal complex (CC).

**Figure 3 antibiotics-10-00413-f003:**
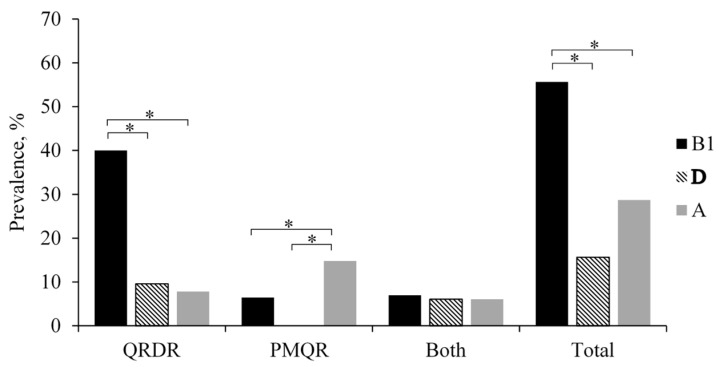
Prevalence of phylogenetic groups in 115 isolates not susceptible to quinolones but harboring mutations in QRDR and/or PMQR genes, from food-producing animals and their food products, collected in the Philippines. * The *p*-value was significant (*p* < 0.05).

**Table 1 antibiotics-10-00413-t001:** Prevalence of *Escherichia coli* in food-producing animals and their food products.

Sample Source	Sample Type	Number of Samples		95% Confidence Interval
		Examined	Positive (%)	
Supermarket	Beef	54	31 (57.4)	0.4321–0.7077
	Chicken	47	41 (87.2)	0.7426–0.9517
	Pork	68	44 (64.7)	0.5217–0.7592
Open market	Beef	52	26 (50.0)	0.3581–0.6419
	Chicken	68	50 (73.5)	0.6143–0.8350
	Pork	48	34 (70.8)	0.5594–0.8305
Abattoir	Beef	28	15 (53.6)	0.3387–0.7249
	Pork	23	11 (47.8)	0.2682–0.6941
	Environmental swab	4	4 (100.0)	0.3976–1.000
Swine farm	Rectal swab	43	29 (67.4)	0.5146–0.8092
	Environmental swab	31	23 (74.2)	0.5539–0.8814
Poultry farm	Cloacal swab	45	24 (53.3)	0.3787–0.6834
	Environmental swab	10	6 (60.0)	0.2624–0.8784
	Egg	50	1 (2.0)	0.0005–0.1065
Buffalo farm	Milk	30	0 (0.0)	0.0000–0.1157
Total tested		601	339 (56.4)	0.5234–0.6041

**Table 2 antibiotics-10-00413-t002:** Antimicrobial resistance profiles of *E. coli* isolates in samples.

Antimicrobial Agents	Sample, *n* (%)							Total (*n* = 791)
	Beef (*n* = 191)	Chicken (*n* = 224)	Pork (*n* = 188)	Egg (*n* = 8)	Cloacal Swabs (*n* = 52)	Rectal Swabs (*n* = 55)	Environmental Swabs (*n* = 73)	
TET	66 (34.6)	148 (66.1)	93 (49.5)	3 (37.5)	40 (76.9)	53 (96.4)	52 (71.2)	455 (57.5)
AMP	63 (33.0)	143 (63.8)	104 (55.3)	7 (87.5)	34 (65.4)	46 (83.6)	51 (69.9)	448 (56.6)
SXT	32 (16.8)	104 (46.4)	87 (46.3)	2 (25.0)	39 (75.0)	37 (67.3)	45 (61.6)	346 (43.7)
CHL	37 (19.4)	86 (38.4)	65 (34.6)	1 (12.5)	16 (30.8)	39 (70.9)	40 (54.8)	284 (35.9)
STR	32 (16.8)	92 (41.1)	49 (26.1)	0	21 (40.4)	19 (34.5)	24 (32.9)	237 (30.0)
NAL	11 (5.8)	101 (45.1)	23 (12.2)	0	43 (82.7)	13 (23.6)	9 (12.3)	200 (25.3)
KAN	9 (4.7)	64 (28.6)	7 (3.7)	0	15 (28.8)	15 (27.3)	6 (8.2)	116 (14.7)
CIP	2 (1.0)	57 (25.4)	6 (3.2)	0	21 (40.4)	3 (5.5)	3 (4.1)	92 (11.6)
CST	13 (6.8)	38 (17.0)	9 (4.8)	2 (25.0)	6 (11.5)	5 (9.1)	5 (6.8)	78 (9.9)
AmC *	6 (6.8) ^a^	12 (10.3) ^b^	10 (11.4) ^c^	0 ^d^	0 ^e^	4 (10.0) ^f^	0 ^g^	32 (8.9)
IPM	3 (1.6)	43 (19.2)	5 (2.7)	0	15 (28.8)	2 (3.6)	2 (2.7)	70 (8.8)
GEN	5 (2.6)	42 (18.8)	11 (5.9)	0	2 (3.8)	2 (3.6)	5 (6.8)	67 (8.5)
FOX	10 (5.2)	28 (12.5)	13 (6.9)	0	9 (17.3)	3 (5.5)	3 (4.1)	66 (8.3)
CTX *	4 (4.5) ^a^	6 (5.1) ^b^	2 (2.3) ^c^	0 ^d^	6 (60.0) ^e^	2 (20.0) ^f^	0 ^g^	20 (5.5)
CAZ	5 (2.6)	9 (4.0)	3 (1.6)	0	3 (5.8)	1 (1.8)	3 (4.1)	24 (3.0)
FEP	3 (1.6)	6 (2.7)	1 (0.5)	0	4 (7.7)	3 (5.5)	1 (1.4)	18 (2.3)
MEM *	0 ^a^	2 (1.7) ^b^	1 (1.1) ^c^	0 ^d^	0 ^e^	0 ^f^	0 ^g^	3 (0.8)

TET—tetracycline; AMP—ampicillin; STX—sulfamethoxazole-trimethoprim; CHL—chloramphenicol; STR—streptomycin; NAL—nalidixic acid; KAN—kanamycin; CIP—ciprofloxacin; CST—colistin; AmC—amoxicillin/clavulanic acid; IPM—imipenem; GEN—gentamicin; FOX—cefoxitin; CTX—cefotaxime; CAZ—ceftazidime; FEP—cefepime; MEM—meropenem. * Only 361 samples were tested with amoxicillin/clavulanic acid, cefotaxime, and meropenem; ^a^ beef = 88; ^b^ chicken = 117; ^c^ egg = 8; ^d^ cloacal swab = 10; ^e^ environmental swab = 40; ^f^ rectal swab = 10; ^g^ pork = 88 samples.

**Table 3 antibiotics-10-00413-t003:** Distribution of multidrug-resistant *E. coli* isolates in samples.

No. of Antimicrobials Classes	Number (%) of Quinolone-Resistant Isolates							
	Beef (*n* = 191)	Chicken (*n* = 224)	Pork (*n* = 188)	Egg (*n* = 8)	Cloacal Swabs (*n* = 52)	Rectal Swabs (*n* = 55)	Environmental Swabs (*n* = 73)	Total (*n* = 791)
0	89 (46.6)	20 (8.9)	51 (27.1)	0 (0.0)	2 (3.8)	1 (1.8)	6 (8.2)	169 (21.4)
1–2	50 (26.2)	52 (23.2)	46 (24.5)	6 (75.0)	5 (9.6)	6 (10.9)	19 (26.0)	184 (23.3)
3–4	38 (19.9)	67 (29.9)	52 (27.7)	2 (25.0)	20 (38.5)	19 (34.5)	27 (37.0)	225 (28.4)
5–6	13 (6.8)	60 (26.8)	37 (19.7)	0	16 (30.8)	25 (45.5)	18 (24.7)	169 (21.4)
7–8	1 (0.5)	24 (10.7)	2 (1.1)	0	8 (15.4)	4 (7.3)	3 (4.1)	42 (5.3)
>9	0	1 (0.4)	0	0	1 (1.9)	0	0	2 (0.3)
Resistance ≥ 1	102 (53.4)	204 (91.1)	137 (72.9)	8 (100.0)	50 (96.2)	54 (98.2)	48 (65.8)	622 (78.6)
MDR ≥ 3	52 (27.2)	152 (67.9) *	91 (48.4)	2 (25.0)	45 (86.5)	48 (87.3)	48 (65.8)	438 (55.4)

MDR: multidrug-resistant *E. coli*. * The *p*-value for beef, chicken, and pork sample data was significant (<0.05).

**Table 4 antibiotics-10-00413-t004:** Distribution of amino-acid substitutions in QRDR genes, PMQR genes, and minimum inhibitory concentration (MIC) of tested quinolones in QRDR/PMQR-harboring *E. coli* isolates.

QRDR Amino-Acid Substitutions ^a^				PMQR	No. of Isolates	MIC (μg/mL) ^d^	
*gyrA*	*gyrB*	*parC*	*parE*			Nalidixic Acid	Ciprofloxacin
Ser83 Leu	-^b^	-^b^	-^b^	-^c^	7	16 – >128	0.125–1
Ser83 Leu	Ser492 Asn	-^b^	-^b^	-^c^	1	>128	0.5
Ser83 Leu	-^b^	Ser80 Ile	-^b^	-^c^	1	>128	2
Asp87 Tyr	-^b^	-^b^	-^b^	-^c^	1	>128	0.5
Ser83 Leu; Asp87 Asn	-	-^b^	-^b^	-^c^	1	>128	4
Ser83 Leu; Asp87 Asn	Ser492 Asn	Ser80 Ile; Glu84 Gly	-^b^	-^c^	4	>128	>32
Ser83 Leu; Asp87 Asn	-^b^	Ser80 Ile	-^b^	-^c^	31	>128	4–>32
Ser83 Leu; Asp87 Asn	-^b^	Ser80 Ile	Ile464 Phe	-^c^	2	>128	>32
Ser83 Leu; Asp87 Asn	-^b^	Ser80 Ile	Ser458 Ala	-^c^	16	128–>128	0.5–>32
Ser83 Leu; Asp87 Asn	-^b^	Ser80 Ile; Glu84 Gly	-^b^	-^c^	2	>128	>32
Ser83 Leu	Ser492 Asn	Ser80 Arg	-^b^	*qnrB4*	1	>128	8
Ser83 Leu	-^b^	Ser80 Ile	-^b^	*qnrS1*	1	>128	32
Ser83 Leu	-^b^	-^b^	-^b^	*qnrS1*	1	>128	4
Ser83 Leu	-^b^	-^b^	-^b^	*qnrB4*	12	>128	1–4
Asp87 Try	-^b^	-^b^	-^b^	*qnrS1*	1	>128	2
Asp87 Try	-^b^	Ser80 Ile	-^b^	*qnrB4*	1	8	<0.03
Ser83 Leu; Asp87 Asn	S492 *n*	S80 I; E84 G	-^b^	*qnrB4*	3	>128	>32
-^b^	S492 *n*	-^b^	-^b^	*qnrB4*	1	16	0.25
-^b^	-^b^	-^b^	-^b^	*qnrS1*	27	8–>128	0.25–2
-^b^	-^b^	-^b^	-^b^	*qnrA1; qnrS1*	1	>128	>32

^a^ QRDR substitutions: *gyrA*—Ser83 Leu: serine to leucine at codon 83; Asp87 Asn: aspartic acid to asparagine/tyrosine at codon 87; *gyrB*—Ser492 Asn: serine to aspartic acid at codon 492; *parC*—Ser80 Ile/Arg: serine to isoleucine/arginine at codon 80; Glu84 Gly: glutamic acid to glycine at codon 84; *parE*—Ser458 Ala: serine to alanine at codon 458; Ile464 Phe: isoleucine to phenylalanine at codon 464. ^b^ No substitution detected in the QRDR. ^c^ No PMQR determinant detected. ^d^ MIC—minimum inhibitory concentration.

## Data Availability

Not applicable.

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
