# Peer review of "Prevalence and Characterization of Quinolone-Resistance Determinants in Escherichia coli Isolated from Food-Producing Animals and Animal-Derived Food in the Philippines"

_antibiotics, 2021, doi:10.3390/antibiotics10040413_

Round 1
Reviewer 1 Report
Belotindos et.al investigated the presence of quinolone-resistant E. Coli and their quinolone-resistant determinants, isolated from food-producing
animals/animal-derived foods in the Philippines. Disc diffusion method detected the resistance of isolates to at least one antimicrobe and multi drugs tested here. Further investigation of Quinolone-resistance determinant region (QRDR) identified amino acid substitution in gyrA, gyrB, parC, and parE genes. Apart from QRDR, plasmid-encoded quinolone resistance genes (PMQR) were also detected. A multilocus sequence (QRDR+PMQR) typing followed by phylogenetic analysis was performed to categorize the isolates into different phylogroups. Authors conclude a high prevalence of E. Coli and chances of the potential spread of quinolone-resistant determinants in food-producing animals and products.
The manuscript is well written and easy to understand. I do not have major comments. Please see the following minor edits required in the manuscript.
Specific comments
Line 224 – Please check the spelling “animals healthly and increase”
Line 297 – Misseplled as “quinolne”
Figure 3 – Misspelled as “prevelence” in the y-axis
Lines 452-466 – Not sure if it is a journal requirement to add ‘Conclusion” after “materials and methods”. If not, please include it after discussion.
Author Response
Reviewer 1:
Thank you for your precious comments. Please find point-to-point resposes to your comments.
Comment 1: Line 224 – Please check the spelling “animals healthly and increase” health.
Response 1: Thank you for your comment. We have revised the text according to your comment.
Comment 2: Line 297 – Misseplled as “quinolne”
Response 2: Thank you for your comment. We have revised the text according to your comment.
Comment 3: Figure 3 – Misspelled as “prevelence” in the y-axis
Response 3: Thank you for your comment. We have revised the text according to your comment. Please find it in Fig. 3.
Comment 4: Lines 452-466 – Not sure if it is a journal requirement to add ‘Conclusion” after “materials and methods”. If not, please include it after discussion.
Response: Thank you for your comment. The “Conclusion” after the “Materials and methods was according to the journal requirement/format.
Reviewer 2 Report
The authors present a study that investigate the prevalence of and characterizes quinolone resistance determinants in Escherichia coli isolates from several sources in the Philippines. According to the authors, this is the first study investigating such molecular characteristics in Escherichia coli from the Philippines. The authors present clear and concise aims, and the results adequately answer these aims. Although somewhat old methodology is used, the reviewer recognizes that whole-genome sequencing infrastructure may not yet be available, but highly recommends considering this for future publications. There are some issues with the English language throughout the manuscript, and the reviewer recommends a thorough language review from the authors.
Reviewer comments:
- While it would make sense to call it “ coli contamination” when discussing food, this is not true for rectal/cloacal/environmental swabs, as it is expected to find E. coli in such niches. The reviewer suggests changing the title of Table 1 and subsection 2.1 accordingly.
- In Table 2, please either use the abbreviated antimicrobial names (all in uppercase letters) or write the full name.
- In Table 3, please write “Environmental” in one word
- Line 113: The title of the section lacks a “Q” in “PMQR”
- Line 127: Exclude the section inside the square brackets and let the Table speak for itself
- Line 130: This is not a phylogenetic tree, but a dendrogram. A phylogenetic tree is based on a nucleic acid/amino acid sequence, where a model of evolution is inferred. This is a clustering technique based on the allele numbers from the MLST results. Please change the name accordingly throughout the manuscript.
- Line 150: “qnr-family genes”, not Qnr
- Line 153: It is unclear whether the authors are talking about the chromosomal genes here, or mutations in chromosomal genes related to PMQR. The reviewer is assuming that the authors mean: “All isolates with PMQR also harboured mutations in at least one QRDR, except for parE”. This is however incorrect when looking at the information presented in Table 4, as 27 isolates only had qnrS1, and one isolate had qnrS1 and qnrA1.
- Line 162: All coli have QRDR present. Please clarify to “harbouring mutations in QRDR and PMQR determinants”. Please make sure to change this throughout the manuscript.
- Line 162: “Sequence types”, not “Single types”
- Figure 2: The significance of the dashed lines is not clarified in the figure text.
- Figure 3: p < 0.05, not 0.5. Also, “Prevalence”, not “Prevelence” on the y-axis.
- Line 224: “Healthy”
- Line 297: Quinolone
- Line 297: As far as the reviewer is aware, no active search for plasmids was conducted in this study, either by PCR, WGS, or S1 nuclease PFGE. Please change the sentence to reflect that.
- Line 301: PMQR
- Line 418: UPGMA, not UPGM
Author Response
Reviewer 2:
Thank you for your precious comments. Please find point-to-point resposes to your comments.
General comment: There are some issues with the English language throughout the manuscript, and the reviewer recommends a thorough language review from the authors.
Response: We already sent and checked the manuscript by English editing company.
Specific comments:
Comment 1: While it would make sense to call it “ coli contamination” when discussing food, this is not true for rectal/cloacal/environmental swabs, as it is expected to find E. coli in such niches. The reviewer suggests changing the title of Table 1 and subsection 2.1 accordingly.
Response 1: Thank you for your suggestion. We have revised the text according to your suggestion.
Comment 2: In Table 2, please either use the abbreviated antimicrobial names (all in uppercase letters) or write the full name.
Response: Thank you for your suggestion. We have revised the antimicrobials names in abbreviated format (all in uppercase letters) according to your suggestion.
Comment 3: In Table 3, please write “Environmental” in one word
Response: Thank you for your comment. We have revised the text according to your comment.
Comment 4: Line 113: The title of the section lacks a “Q” in “PMQR”
Response: Thank you for your comment. We have revised the text according to your comment.
Comment 5: Line 127: Exclude the section inside the square brackets and let the Table speak for itself
Response: Thank you for your suggestion. We have revised the text according to your suggestion.
Comment 6: Line 130: This is not a phylogenetic tree, but a dendrogram. A phylogenetic tree is based on a nucleic acid/amino acid sequence, where a model of evolution is inferred. This is a clustering technique based on the allele numbers from the MLST results. Please change the name accordingly throughout the manuscript.
Response: Thank you for your suggestion. We have revised the text according to your suggestion.
Comment 7: Line 150: “qnr-family genes”, not Qnr
Response: Thank you for your comment. We have revised the text according to your comment.
Comment 8: Line 153: It is unclear whether the authors are talking about the chromosomal genes here, or mutations in chromosomal genes related to PMQR. The reviewer is assuming that the authors mean: “All isolates with PMQR also harboured mutations in at least one QRDR, except for parE”. This is however incorrect when looking at the information presented in Table 4, as 27 isolates only had qnrS1, and one isolate had qnrS1 and qnrA1.
Response: Thank you for your comment. We have revised the statement according to your comment.
Comment 9: Line 162: All coli have QRDR present. Please clarify to “harbouring mutations in QRDR and PMQR determinants”. Please make sure to change this throughout the manuscript.
Response: Thank you for your comment. We have revised the text according to your comment.
Comment 10: Line 162: “Sequence types”, not “Single types”
Response: Thank you for your comment. We have revised the text according to your comment.
Comment 11: Figure 2: The significance of the dashed lines is not clarified in the figure text.
Response: Thank you for your comment. We have revised the figure text and added labeled information on the figure to clarify the significance of the dashed lines.
Comment 12: Figure 3: p < 0.05, not 0.5. Also, “Prevalence”, not “Prevelence” on the y-axis.
Response: Thank you for your comment. We have revised the text according to your comment.
Comment 13: Line 224: “Healthy”
Response: Thank you for your comment. We have revised the text according to your comment.
Comment 14: Line 297: Quinolone
Response: Thank you for your comment. We have revised the text according to your comment.
Comment 15: Line 297: As far as the reviewer is aware, no active search for plasmids was conducted in this study, either by PCR, WGS, or S1 nuclease PFGE. Please change the sentence to reflect that.
Response: Thank you for your comment. We have revised the text reflecting we did not conduct plasmid analysis either by PCC, WGS or S1 nuclease PFGE.
Comment 16: Line 301: PMQR
Response: Thank you for your comment. We have revised the text according to your comment.
Comment 17: Line 418: UPGMA, not UPGM
Response: Thank you for your comment. We have revised the text according to your comment.
Reviewer 3 Report
The manuscript by Belotindos et al aims at investigating the quinolone resistance determinants in E. coli isolated in the Philippines from food-producing animals and their food products. Fluoroquinolones are critical and high-priority drugs for humans, the use of which in food producing animals has been significantly teduced or banned already in many countries, but in the Philippines a significant percentage of infection cases are still caused by animal-derived food because of poor hygiene and unproper inspections.
Overall the ms is interesting and well-written.
I have only few comments:
1) How did you select the 141 E. coli isolates non susceptible to quinolones? According to Table 2 you had 292 of such isolates. Correct? Which criterion did you follow for chosing the 141 isolates that you studied at the molecular level?
2) Though very informative, I suggest to move Figure 1 in Supplementary Material
3) It would be nice to have a figure of disk diffusion assay of representative isolates belonging to ST155, ST162, ST354, ST359 tested for many antibiotics. Typically this kind of figure is quite impactful.
4) Page 2, Line 42: "difficult to treat after individuals succumb to them" does not read correct.
5) Page 10, line 336-339: please rephase.
Author Response
Reviewer 3:
Thank you for your precious comments. Please find point-to-point resposes to your comments.
Comment 1: How did you select the 141 E. coli isolates non susceptible to quinolones? According to Table 2 you had 292 of such isolates. Correct? Which criterion did you follow for choosing the 141 isolates that you studied at the molecular level?
Response: Thank you for your comment. The criterion we follow in selecting the 141 isolates subjected for molecular analysis was according to disk diffusion method and E. coli isolates that had intermediate and/or resistant to nalidixic acid and/or ciprofloxacin, were selected by screening for quinolone resistance determinants.
Comment 2: Though very informative, I suggest to move Figure 1 in Supplementary Material
Response: Thank you very much for your suggestion. We would like to retain Figure 1 in the main text which we believe it shows important information regarding of the tested samples such quinolone resistance determinant, sequence type, and disk diffusion assay results.
Comment 3: It would be nice to have a figure of disk diffusion assay of representative isolates belonging to ST155, ST162, ST354, ST359 tested for many antibiotics. Typically this kind of figure is quite impactful.
Response: Thank you for your suggestion. However, we cannot able to provide additional figure because we have already presented the diffusion assay data in the last column of the Figure 1, including isolates belonging to ST155, ST162, ST354, ST359 tested for 17 antibiotics.
Comment 4: Page 2, Line 42: "difficult to treat after individuals succumb to them" does not read correct.
Response: Thank you for your comment. We have revised the text according to your comment.
Comment 5: Page 10, line 336-339: please rephase.
Response: Thank you for your comment. We have revised the text according to your comment.